# Coffee Increases Post-Exercise Muscle Glycogen Recovery in Endurance Athletes: A Randomized Clinical Trial

**DOI:** 10.3390/nu13103335

**Published:** 2021-09-23

**Authors:** Laís Monteiro Rodrigues Loureiro, Eugênio dos Santos Neto, Guilherme Eckhardt Molina, Angélica Amorim Amato, Sandra Fernandes Arruda, Caio Eduardo Gonçalves Reis, Teresa Helena Macedo da Costa

**Affiliations:** 1Health Sciences Graduate Program, Nutritional Biochemistry Laboratory, Universidade de Brasília, Brasilia 70910-900, Brazil; laismrloureiro@gmail.com; 2Health Sciences Graduate Program, Faculty of Health Sciences and Faculty of Medicine, Universidade de Brasilia, Brasilia 70910-900, Brazil; eugeniosneto@unb.br; 3Exercise Physiology Laboratory, Faculty of Physical Education, Universidade de Brasilia, Brasilia 70910-900, Brazil; gmolina@unb.br; 4Molecular Pharmacology Laboratory, Department of Pharmaceutical Sciences, Faculty of Health Sciences, Universidade de Brasília, Brasilia 70910-900, Brazil; angelicamato@unb.br; 5Nutritional Biochemistry Laboratory, Department of Nutrition, Universidade de Brasília, Brasilia 70910-900, Brazil; sandrafarruda@gmail.com (S.F.A.); caioedureis@gmail.com (C.E.G.R.)

**Keywords:** coffee, caffeine, glycogen, glucose, insulin, recovery, endurance training, sports nutrition

## Abstract

Coffee is one of the most widely consumed beverages worldwide and caffeine is known to improve performance in physical exercise. Some substances in coffee have a positive effect on glucose metabolism and are promising for post-exercise muscle glycogen recovery. We investigated the effect of a coffee beverage after exhaustive exercise on muscle glycogen resynthesis, glycogen synthase activity and glycemic and insulinemic response in a double-blind, crossover, randomized clinical trial. Fourteen endurance-trained men performed an exhaustive cycle ergometer exercise to deplete muscle glycogen. The following morning, participants completed a second cycling protocol followed by a 4-h recovery, during which they received either test beverage (coffee + milk) or control (milk) and a breakfast meal, with a simple randomization. Blood samples and muscle biopsies were collected at the beginning and by the end of recovery. Eleven participants were included in data analysis (age: 39.0 ± 6.0 years; BMI: 24.0 ± 2.3 kg/m^2^; VO_2max_: 59.9 ± 8.3 mL·kg^−1^·min^−1^; PPO: 346 ± 39 W). The consumption of coffee + milk resulted in greater muscle glycogen recovery (102.56 ± 18.75 vs. 40.54 ± 18.74 mmol·kg dw^−1^; *p* = 0.01; d = 0.94) and greater glucose (*p* = 0.02; d = 0.83) and insulin (*p* = 0.03; d = 0.76) total area under the curve compared with control. The addition of coffee to a beverage with adequate amounts of carbohydrates increased muscle glycogen resynthesis and the glycemic and insulinemic response during the 4-h recovery after exhaustive cycling exercise.

## 1. Introduction

Coffee is one of the most widely consumed beverages in the world. Its beneficial effect on endurance sports, such as cycling and triathlon, is related to the role of caffeine in improving performance, reducing the perception of effort, fatigue, or pain associated with exercise, when consumed before training [1,2]. Beyond sports, coffee also has beneficial effects on glucose metabolism, especially when caffeinated coffee is consumed on a long-term basis [3]. Data from experimental studies assessing acute caffeinated and decaffeinated coffee consumption indicate increased insulin secretion and insulin sensitivity [4,5,6].

Carbohydrates are the primary energetic substrate for physical exercise and determine athletes’ performance during moderate to high endurance sports [7,8,9]. Although muscle glycogen stores do not completely recover after a few hours of an exercise session, it is important that athletes follow nutritional recommendations to maximize their glycogen synthesis rate, focusing on the early phase of recovery (0–4 h) because of the slightly higher synthesis rates during this period [10]. Special attention to the latter recommendation should be given when training more than once daily or in competitions in which athletes undergo multiple and sequential bouts/rounds [8].

Coffee bioactive compounds such as caffeine, caffeic acid, and cafestol have been shown to improve glucose metabolism and promote post-exercise glycogen resynthesis when consumed during recovery [11,12]. However, no study has been conducted to address whether coffee consumption with carbohydrates impacts post-exercise muscle glycogen recovery.

The selection of appropriate foods and/or beverages for post-exercise meals should also consider other nutritional goals related to recovery, such as rehydration and muscle protein synthesis [8]. In this regard, milk is an adequate option, as in addition to the potential for rehydration due to the presence of water and minerals, it also contains carbohydrates and proteins of high digestibility and biological value [13]. Moreover, a study conducted with data from the Brazilian National Dietary Survey collected from a probabilistic sample showed that the coffee brewing and preparation method most frequently used by Brazilians was filtered coffee and coffee with milk [14].

Coffee consumption has well-established effects on insulin sensitivity and glucose homeostasis, but little is known about its effects on glucose metabolism when consumed after exercise. We hypothesized that the combination of coffee with the recommended amounts of carbohydrates favor post-exercise muscle glycogen resynthesis following an exercise session. To test this hypothesis, we investigated the effect of coffee consumption in combination with sweetened milk and a solid meal on post-exercise muscle glycogen resynthesis in endurance athletes.

## 2. Materials and Methods

This study was a randomized, double-blind, crossover clinical trial conducted at the Nutrition Biochemistry Laboratory at the University of Brasília. After undergoing a preliminary incremental cardiopulmonary test, participants were assessed on four occasions. The evening before each session they performed an exhaustive cycle ergometer exercise until volitional fatigue to deplete muscle glycogen stores. Then they followed a low-carbohydrate dietary prescription for the evening meal. The next morning, they returned to the laboratory after a 10–12 h overnight fast to cycle again until volitional fatigue, and started the 4-h recovery period, during which they were provided with one of the two beverage options (coffee + milk or milk) and a breakfast meal, with a simple randomization. Blood samples were collected at regular intervals and muscle biopsies were performed immediately after exercise and at the end of 4-h recovery to determine total muscle glycogen resynthesis (Figure 1).

The sample size was based on the randomized distribution of the study participants in two sequential tests. Muscle glycogen content was the main outcome variable and we considered data from Pedersen et al., (2008), whose intervention with carbohydrates + caffeine after exercise provided a greater post-exercise muscle glycogen resynthesis when compared with carbohydrates alone (*n* = 7; 313 ± 69 mmol·kg dw^−1^ vs. 234 ± 50 mmol·kg dw^−1^, respectively) [15]. Using G Power version 3.1.9.2 (University of Düsseldorf, Germany), the minimal sample size required for a 95% statistical power and an alpha error of 5% (two-tails) resulted in 11 participants. To allow for dropping out and to avoid new recruitment, the sample size was increased by 25%, totaling 14 participants. Recruitment began in August 2018 and continued until the end of data collection in June 2019, when the sample was complete.

We selected 14 healthy endurance-trained adult men, cyclists (*n* = 11; 79%) and triathletes (*n* = 3; 21%), with at least 60 km of cycling per week (4 h/wk), and one year of experience in the sport [16]. The study was disclosed via contact with coaches and sport nutritionists, and a digital folder on social media. All participants should be regular coffee and milk consumers, without allergy, intolerance or discomfort related to the consumption of milk and dairy products. Those who had a daily caffeine ingestion greater than 500 mg, with any medical condition or recent injury, smokers, users of illicit drugs and those undergoing medical treatment were not selected.

After interview, screening, weight and height measurement, and body composition assessment, the participants performed a maximal incremental cardiopulmonary exercise test in the Exercise Physiology Laboratory at the Faculty of Physical Education, University of Brasília. The test was performed on a mechanically braked cycle ergometer-BIOTEC 1800 (CEFISE Biotecnologia ME, São Paulo, SP, Brazil), and maximal oxygen uptake (VO_2max_) was measured by pulmonary gas exchange using the Ergoespirometer-Cortex Metalyzer 3B (Biophysik, Leipzing, Germany) at a controlled temperature (21–24 °C) and relative humidity of 50–60%.

The last complete minute of maximal incremental cardiopulmonary exercise test was considered for obtaining the VO_2max_ and the peak power output (PPO). Participants were included in the next experiment phase if they met the requirements of VO_2max_ ≥ 45 mL·kg^−1^·min^−1^ and PPO 280 watts (W) [16]. The PPO achieved in this test was used for prescribing the muscle glycogen depletion exercise protocol performed during the two experimental sessions.

In the preparatory phase of each experimental session muscle glycogen levels were depleted by exhaustive cycle exercise and dietary intervention. About 15 h before the session, participants attended the Nutrition Biochemistry Laboratory at the University of Brasília to perform an interval exercise until fatigue on a cycle ergometer CG-04 (Inbramed, Porto Alegre, RS, Brazil) at a controlled temperature (21–24 °C) and relative humidity of 50–60%. The protocol [17] started with a 5 min warm-up with 20% loading of the PPO at a rate of 60 revolutions per minute (rpm). The load was then increased to 90% PPO for 2 min, followed by 2 min of recovery at 50% PPO. This attack and recovery protocol was maintained until participants could no longer maintain 60 rpm for 15 s. At this time, the attack phase was reduced to 80% PPO for 2 min, with 2 min of recovery at 50% PPO. When the cadence was reduced again, the attack phase went to 70% and then 60%. Thus, when the participant could no longer complete the 2 min at 60% PPO, the protocol was terminated with 5 min of deceleration at 20% PPO. They received two verbal stimuli whenever the cadence was reduced and received no information about the loads used, the number of attack and recovery cycles, and the total exercise time.

On the morning of the experiment, fasting capillary glucose levels were measured (Accu-Check Performa; Roche Diabetes Care, São Paulo, SP, Brazil) and the leg in which the biopsy procedure would be performed was randomly defined. The first blood sample was taken from the opposite arm with a hypodermic needle and a tube with coagulation activator and separating gel (Greiner Bio-One, São Paulo, SP, Brazil).

After 10 min of sitting rest, the second exercise protocol was initiated to guarantee maximum muscle glycogen depletion. The protocol started with a 5 min warm-up with a 20% PPO workload and a 60 rpm cadence, followed by an increase to 70% PPO workload, in a seated position, until volitional fatigue, when cadence was below 60 rpm for more than 15 s, even after two verbal stimuli. At this point, the workload was reduced to 20% PPO for 5 min of deceleration. Participants consumed water ad libitum and received no information about the loads used and the total exercise time.

The 4-h recovery period began after the exercise. Participants were transferred to a hospital bed in a horizontal supine position. An anesthetic ointment (lidocaine hydrochloride 2%) was applied at the biopsy site and the second blood collection was performed with a catheter positioned in the antecubital region of the arm (Greiner Bio-One, São Paulo, SP, Brazil) that was removed only at the end of the recovery. The catheter was then flushed with sterile saline (Equiplex, Aparecida de Goiânia, GO, Brazil), a procedure that was repeated after each subsequent blood draw. In the first experimental session, biopsies were performed in a leg randomly defined and in the second experimental session, the procedure was performed on the opposite leg. After intramuscular anesthesia (lidocaine hydrochloride 2%), the first biopsy (b1) was taken from the bulkier portion of the contracted vastus lateralis with a Bergström needle modified with suction using a 60 mL syringe. Approximately 150 mg of muscle tissue was removed, trimmed, aliquoted, immediately frozen in liquid nitrogen and stored at −80 °C. After the procedure, the volunteer received three doses of test beverages (0, 60 and 120 min). After 4 h of recovery, a second biopsy was performed (b2) on the same leg, about 5 cm distal from b1. Blood samples were drawn at regular intervals after the first beverage dose (30, 60, 90, 120, 180 and 240 min) throughout recovery (Figure 2). The conditions remained constant for all procedures (21–24 °C, relative humidity of 50–60%, internal diurnal light). The sessions lasted 6 h and the interval between each session was 7–14 days.

Participants received a frappe-type drink prepared with filtered coffee, skimmed milk and sucrose (coffee + milk), or a milk-shake-type drink prepared with water, skimmed milk and sucrose (milk) at 0 and 60 min of the recovery period. At 120 min they received a meal with a salted egg and cottage cheese sandwich, accompanied by filtered coffee with sucrose (coffee + milk) or water with sucrose (milk). Each beverage and the meal were planned to provide 1.2 g of carbohydrates·kg^−1^ and 0.3 g of proteins·kg^−1^, corresponding to the 4:1 ratio of carbohydrates: proteins. The total caffeine provided with the three doses of coffee + milk treatment was 8 mg.kg^−1^.

To prepare the beverages and sandwiches, the ingredients were weighed according to the participants body mass. A content of 1530 mg of caffeine per 100 g of coffee powder was obtained from the analysis of the coffee (Café Torrado e Moído Melitta Tradicional^®^) by HPLC, in the Laboratory of Nutritional and Food Biochemistry, of the Department of Chemistry of the Federal University of Rio de Janeiro. Coffee preparation was carried out using the 10% filtering method with boiling water in a filter paper (Melitta^®^ 102), considering a 50% retention of caffeine in the filter [18] and respecting the ratio of 10 g of powder to 100 mL of water. The filtered coffee was poured into ice cube tray and then frozen, as well as the water. In the second stage of preparation, the beverages were homogenized. Powdered milk (Leite em Pó Desnatado Instantâneo Piracanjuba^®^) and sugar (Cristalçúcar União^®^) were blended for thirty seconds with cold mineral water. Then the frozen coffee (coffee + milk) or water (milk) cubes were added. The beverages were homogenized for two more minutes and served in lidded cups. The sandwich was prepared with white bread (Pão de Sanduíche Forma Seven Boys^®^), salted egg and cottage cheese (Queijo tipo Cottage Canto de Minas^®^). It was accompanied by filtered coffee with sucrose (coffee + milk) or water with sucrose (milk). Beverage volume and sandwich weight were recorded in spreadsheets. The detailed recipe of the beverage is available in Teixeira et al., (2020) [19].

To blind the participants, they were told there were five beverage possibilities (water with sugar, coffee with milk and sugar, milk with sugar, decaf coffee with sugar, and coffee with sugar), and they would be assigned to receive two of these options in sessions 1 and 2. In addition, beverages were served in an opaque lidded cup with a dark straw. Finally, to mask the smell, coffee was prepared every day in the lab, regardless of whether it was a test or control session. The lids of the cups were soaked with coffee to give it the drink smell. Researchers conducting the exercise depletion, blood collection and biopsy, and those assessing the outcomes were also blinded to the intervention assignment until the end of the data analysis. The researchers who generated the random allocation sequence also assigned participants to interventions and provided the treatments.

To identify any caffeinated products regularly consumed by the participants, they answered a questionnaire including information about medication and dietary supplement use, the type and frequency of consumption, with special attention to caffeine, in addition to a quantitative food frequency questionnaire with caffeine sources [20]. They were all regular coffee consumers, and the individual daily caffeine intake did not exceed the limit of 500 mg (mean caffeine intake of 296 ± 111 mg).

During the study, the participants were instructed to maintain their usual exercise routine and eating habits. The exception was the recommendation to completely exclude supplements during the study period, as well as alcoholic beverages, caffeine-source foods and medications for two days prior to each experimental session. 

After the depleting exercise protocol, they received a low-carbohydrate dietary prescription (0.8 g of carbohydrates·kg^−1^; 0.8 g of proteins·kg^−1^; 1.0 g of lipids·kg^−1^) with options to be consumed until 9 p.m. They returned the next morning at 7 a.m., after a 10–12 h fast, for the experimental session at the same laboratory. In each experimental session an evening meal recall was performed and analyzed on NDSR Software (Nutrition Data System for Research, Nutrition Coordinating Center, University of Minnesota, MN, USA) to verify compliance to the low-carbohydrate evening dietary prescription. Water consumption was ad libitum at all stages. Participants were asked to repeat their training and feeding routine in the days prior to the return for the second session.

To quantify glycogen in the muscle tissue, we used an adapted method described by Lo et al., (1970) [21]. One aliquot (35–50 mg) of muscle tissue was incubated with 500 μL of 30% potassium hydroxide solution saturated with sodium sulfate at 100 °C for 30 min. A 95% Ethanol (1.2 volumes) was added, and the samples were incubated on ice for 30 min to precipitate glycogen from alkaline digestion. The glycogen precipitate was diluted five times and 200 μL of 5% phenol and 1 mL of 98% sulfuric acid was added. The absorbance was recorded at 490 nm in a spectrophotometer (Libra, Biochrom, Cambrige, England). The final concentration of glycogen was obtained based on a standard curve range from 0–100 mg·mL^−1^. The final tissue glycogen concentration was expressed in mmol·kg dw^−1^ following the proposed conversion of Areta & Hopkins (2018) [22].

To determine muscle glycogen synthase activity, we used the spectrophotometric method described by Kornfeld & Brown (1962) [23]. One aliquot of tissue (50 mg) was homogenized with 800 μL of buffer solution (Tris-EDTA) and a protease inhibitor cocktail. After centrifugation at 10,000 g, 4 °C, 15 min, one aliquot of the homogenate (200 μL) was added to a cocktail containing UDP-glucose, glycogen, and glucose-6-phosphate, and incubated at 30 °C for 60 min. The reaction was stopped by boiling for 5 minutes, and the sample was centrifuged at 10,000 g, 4 °C, 2 min. Then, 25 μL of the supernatant was added to a second cocktail containing phosphoenolpyruvate and NADH, in addition to a pyruvate kinase and lactate dehydrogenase solution. The absorbance was monitored at 340 nm during 5 min in a spectrophotometer (Libra, Biochrom, Cambrige, England) to determine the consumption of 1 nmol of NADH by each unit of glycogen synthase. A total blank was performed with water instead of the sample, and a specific blank was performed for each sample with boiled homogenized tissue. Total protein was determined by the method of Hartree (1972) [24] and the enzyme activity was expressed in units of glycogen synthase per mg of protein (U·mg protein^−1^).

Venous blood samples were maintained at room temperature for 15 min after collection, and then centrifuged (1792 g, 4 °C, 15 min) and cooled to 2 °C. The separated blood-fraction tube was transported in a cool-bag to the accredited Analytical Laboratory for analysis. Serum glucose and insulin levels were determined by enzymatic and chemiluminescence methods, respectively. The total area under the curve (TAUC) of glucose and insulin was calculated using GraphPad Prism, version 6 (GraphPad Software Inc., San Diego, CA, USA).

Personal individual characteristics, metabolic parameters and macronutrient intake were presented as means and standard deviations (SD). Repeated measure effect analysis using the mixed models procedure (PROC MIXED), which takes into account the covariance structure of the data, was implemented on SAS/STAT^®^ software (SAS on demand, SAS Institute Inc., Cary, NC, USA). The model protocol in PROC MIXED used the REML (restricted maximum likelihood) option with sequence, subjects, order and treatment included. Each subject followed a sequence of Milk or Coffee + milk in randomly defined order from the first session. The sequence and order protocol were considered for all performed analysis, considering the influence of the first experimental protocol on the subject depletion and response to treatment, subsequently. The least-square means (LS means) values and their standard error (SE) of SAS PROC MIXED were presented and compared between coffee + milk and milk treatments using the Student’s *t*-test at a significance level of 0.05. The null hypothesis was the equality of the means values between the two treatments, and the alternative hypothesis was the significant difference of the means. Cohen-d effect size (ES) was calculated to measure the magnitude of the difference between milk and coffee + milk. Thresholds for small, moderate, and large effects were 0.20, 0.50, and 0.80 [25].

The study was approved by the Research Ethics Committee of the Faculty of Health Sciences of the University of Brasilia (appraisal no. 1.657.099), Brazil. The study was conducted in accordance with Helsinki Declaration and is registered in The Brazilian Clinical Trials Registry (ReBEC) under number RBR-9nrxdf (https://ensaiosclinicos.gov.br/rg/RBR-9nrxdf) (Access in 23 September 2020). All participants were informed about the purpose and risks related to the study and signed the informed consent form before enrolment.

## 3. Results

### 3.1. Participants

Fourteen participants completed both experimental sessions (Figure 3). Three of them had gastrointestinal symptoms after consuming the beverages and meals, including abdominal discomfort and diarrhea and were, therefore, excluded. Eleven participants were included in data analysis with the following characteristics: age 39.0 ± 6.0 years, body mass 75.7 ± 7.6 kg, body mass index 24.0 ± 2.3 kg/m^2^, body fat 16.3 ± 5.4%, VO_2max_ 59.97 ± 8.28 mL·kg^−1^·min^−1^, PPO 346 ± 39 W. Total exercise time and macronutrient intake in the evening meal did not differ between the two experimental sessions (Table 1), showing that participants followed the dietary and exercise protocols for depleting muscle glycogen equally at both sessions. The total 1st and 2nd beverage volume was 530 ± 54 mL. The 3rd beverage volume (coffee or water) and breakfast meal weight consumed were 275 ± 46 mL and 175 ± 12 g, respectively.

### 3.2. Muscle Glycogen Concentration

At the end of the exercise performed on the morning of the experimental session, muscle glycogen levels (b1) were depleted, and no difference was observed between the two sessions (103.85 ± 24.15 vs. 120.75 ± 24.15 mmol·kg dw^−1^ for milk and coffee + milk, respectively; *p* = 0.48). After the 4-h recovery period, the consumption of coffee + milk resulted in greater glycogen accumulation than milk, with a difference between final and initial concentrations (Δ) 153% greater for coffee + milk (102.56 ± 18.75 mmol·kg dw^−1^) compared to Milk (40.54 ± 18.74 mmol.kg dw^−1^) (*p* = 0.01), resulting in a large effect size (d = 0.94) for coffee + milk (Figure 4a).

### 3.3. Glycogen Synthase Activity

At the end of the recovery period, a decrease in glycogen synthase activity was observed with both treatments. This decrease was 149% greater after the consumption of coffee + milk than after the consumption of Milk. However, the difference was not statistically significant (−9.10 ± 13.52 vs. −35.12 ± 13.59 U·mg protein^−1^ for milk and coffee + milk, respectively, *p* = 0.22) (Figure 4b).

### 3.4. Serum Glucose and Insulin Levels

Fasting serum glucose and insulin levels did not differ between the two experimental sessions [glucose: 4.63 ± 0.10 vs. 4.63 ± 0.10 mmol. L^−1^ for milk and coffee + milk, respectively (*p* = 0.98); insulin: 371.32 ± 99.30 vs. 434.03 ± 99.30 pmol. L^−1^ for milk and coffee + milk, respectively (*p* = 0.84)]. Analysis of glucose and insulin TAUC indicated that coffee consumption increased glycemic (*p* = 0.02; d = 0.83) and insulinemic response (*p* = 0.03; d = 0.76) (Figure 5).

## 4. Discussion

In this study, we showed that coffee increased post-exercise muscle glycogen resynthesis and the glycemic and insulinemic response during a 4-h recovery period after an exhaustive cycling exercise session, when compared with the control. This is consistent with our hypothesis that the combination of coffee with the recommended amounts of carbohydrates would favor post-exercise muscle glycogen resynthesis.

In addition to the provision of mixed carbohydrates sources in the beverages, milk is rich in amino acids that promote increased insulin secretion, enhancing the muscle glucose uptake [26]. Milk was consumed by the participants at the intervention (coffee + milk) and control (milk) settings, therefore, the higher glycemic and insulinemic response following coffee + milk consumption can be attributed to the effect of coffee on glucose metabolism. A systematic review that analyzed the effects of coffee consumption on glucose metabolism reported an increase in the area under the curve in the glycemic response in the first hours after consumption of caffeinated coffee [3]. Our group investigated whether the incretin hormones were related to this outcome through increased intestinal absorption of carbohydrates; however, results did not confirm this hypothesis [4]. In addition, a meta-analysis showed a reduction in insulin sensitivity in healthy individuals after acute caffeine intake [27]. In light of these findings, we presume that caffeine was one of the components in coffee increasing the glycemic and insulinemic response reported herein.

The effect of coffee consumption on carbohydrate metabolism in our experiment must be considered in light of the effect of exercise. The action of coffee, more precisely of caffeine, to induce insulin resistance [27] did not impair the significant muscle glycogen accumulation during the recovery period. Physical exercise increases glucose uptake in an insulin-independent manner, especially during the early recovery, due to exercise-related depletion of muscle glycogen, greater release of Ca^2+^ from sarcoplasmic reticulum in muscle cells and the consequent translocation of GLUT-4 transporters to the cell membrane [7,28,29]. In this sense, it is important to highlight that muscle glycogen resynthesis is the result of several simultaneous effects of coffee consumption and exercise itself in glucose metabolism. Moreover, exercise [30] and other coffee constituents [4] can reduce the effect of caffeine on insulin sensitivity. The mechanisms through which coffee acts on energy homeostasis in the setting of exercise have not been fully elucidated, as this is a recent research topic. However, it is currently known that coffee components, such as caffeine, cafestol and caffeic acid may play an important role in muscle glycogen recovery [11].

To explore the effect of coffee consumption on post-exercise muscle glycogen resynthesis during recovery in detail, we assessed the activity of glycogen synthase, the enzyme that catalyzes the incorporation of glycosyl residues from glucose into glycogen [7]. We observed a decrease in glycogen synthase activity in both treatments and the reduction was greater after the consumption of coffee + milk, although the result was not statistically significant. Glycogen synthase activity is stimulated by low muscle glycogen and high glucose-6-phosphate content, in addition to the presence of insulin [31,32]. Considering the increase in glucose and insulin levels in response to the consumption of coffee, we suggest that coffee promoted greater muscle glucose uptake, with earlier activation of glycogen synthase and consequently earlier and greater glycogen resynthesis during the recovery period. As the high glycogen concentration inhibits the enzyme activation [31], the earlier and greater glycogen resynthesis was followed by the inhibition of glycogen synthase activity at the end of the recovery period (240 min). On the other hand, milk treatment presumably resulted in later glycogen synthase maximal activity during recovery, which is consistent with the lower post-exercise muscle glycogen accumulation. Therefore, we cannot state a cause-and-effect relationship between coffee consumption and the reduction in glycogen synthase activity. Rather, the effect of coffee is more likely to have been due to an early increase of enzyme activity, leading to greater glycogen accumulation. Similar findings were reported by Battram et al., (2004), who investigated the effects of caffeine intake on glycogen resynthesis in healthy men after glycogen-depleting exercise. The authors reported that caffeine intake was associated with a significant decrease in glycogen synthase activity at the end of the 5–h recovery period [33]. In our study, the high interpersonal variability in glycogen synthase enzymatic activity among the participants may explain the non-significant difference between treatments. Furthermore, a muscle biopsy performed at the mid-recovery time could have shown the increase in glycogen synthase activity promoted by coffee consumption before the reduction in activity promoted by the accumulation of muscle glycogen. It is important to point that we did not estimate sample size using enzyme activity variability data. Therefore, a larger sample size should be considered in future trials to define the effects of coffee consumption on glycogen synthase activity.

There is a large variability in muscle glycogen content reported by different studies due to the level of athletes’ training, type, frequency and type of supplements administration, and the total recovery period [15,22]. To ensure maximum depletion of muscle glycogen in this study, the participants performed two blocks of exhaustive exercise separated by a low-carbohydrate meal and an overnight fasting. This protocol warranted muscle glycogen depletion with exercise and dietary intervention [34], and liver glycogen depletion with fasting [35].

We acknowledge that hepatic glycogen depletion protocol usually employed in studies with athletes does not correspond to real life training, since high intensity exercise training while fasting does not improve performance [36]. However, fasting is a necessary standardization to decrease interpersonal variations of fuel status. We can infer that part of the carbohydrates supplied by the beverages were metabolized in the liver (galactose and fructose) and directed towards hepatic glycogen resynthesis [37,38]. This aspect is especially important due to the high degree of hepatic glycogen depletion to which participants were subjected. The provision of a carbohydrate-rich meal from mixed sources (glucose, fructose and galactose) favors and accelerates the intestinal uptake of monosaccharides compared with glucose alone, by enabling the use of different transporters in enterocytes [39]. Mixed-carbohydrates source does not necessarily promote an increased rate of muscle glycogen recovery [37,40,41]. However, it increases the feasibility of large amounts of carbohydrate intake by athletes, since gastrointestinal intolerance is minimized [41]. This, in turn, favors muscle glycogen resynthesis.

Despite all participants having reported habitual consumption of coffee and milk, without allergy, intolerance and discomfort, three of them experienced abdominal discomfort and diarrhea. Those with higher body mass received higher volumes of the beverages, and consequently a high load of lactose and sucrose. Even those who had never experienced gastrointestinal symptoms related to milk consumption may have experienced symptoms due to this higher load. Approaches to overcome the gastrointestinal distress related to milk consumption is the combination of lactase enzyme, which aids lactose digestion when this enzyme is the limiting factor [42]. Another possibility is to reduce the sucrose content to obtain a total of 0.8–1.0 g of carbohydrates·kg^−1^·h^−1^, without affecting the efficiency of muscle glycogen recovery [43,44].

Blinding is an inherent limitation of our study, since the two beverages (coffee + milk and milk) have different flavors. In an attempt to reduce this limitation, we have used strategies to blind participants to the aroma and the beverage they would receive in the second experimental session, ensuring the double-blind design of this study.

This is the first study that investigated the effect of a coffee beverage after exhaustive exercise on muscle glycogen resynthesis, glycogen synthase activity and glycemic and insulinemic response in endurance athletes. A great advantage of the beverage recipe prepared in this trial was the possibility to control the carbohydrate content by increasing or reducing the amount of sugar, according to the athlete’s need [19]. Moreover, the beverage was made with simple and easily accessible ingredients and could be prepared at home by any athlete.

## 5. Conclusions

The consumption of coffee with sweetened milk improved the muscle glycogen resynthesis during the 4-h recovery period after the exhaustive cycling exercise when compared to the consumption of sweetened milk. The addition of coffee to a post-exercise beverage with adequate amounts of carbohydrates is an effective strategy to improve muscle glycogen recovery for those cycling athletes with a short-time recovery (<4 h) or competitions with multiple and sequential bouts of exercise. It is currently not possible to define the exact coffee components underlying the latter effect. However, caffeine, caffeic acid, and cafestol are plausible candidates, given previous findings indicating their effect on insulin secretion and muscle glucose uptake. These findings add to the current knowledge on the ergogenic properties of coffee.

## Figures and Tables

**Figure 1 nutrients-13-03335-f001:**
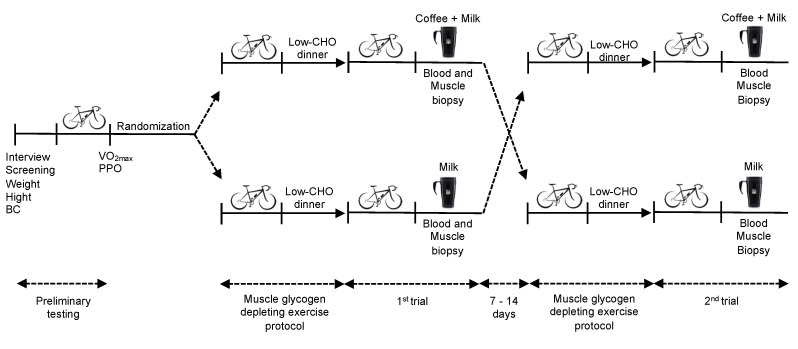
Overview of the experimental trial. BC: body composition; CHO: carbohydrates; PPO: peak power output; VO_2max_: maximal oxygen uptake.

**Figure 2 nutrients-13-03335-f002:**
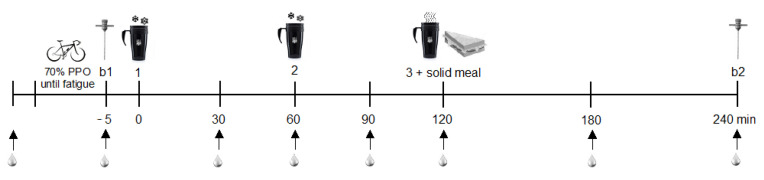
Experimental session time course. b1: first biopsy; b2: second biopsy; PPO: peak power output; 1: first beverage dose; 2: second beverage dose; 3: third beverage dose.

**Figure 3 nutrients-13-03335-f003:**
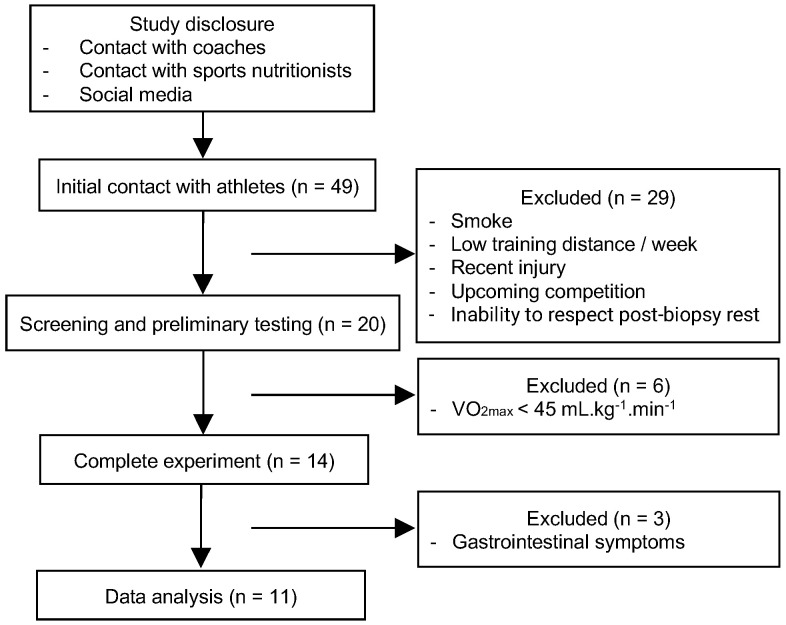
Flow diagram of the progress through the phases of the study.

**Figure 4 nutrients-13-03335-f004:**
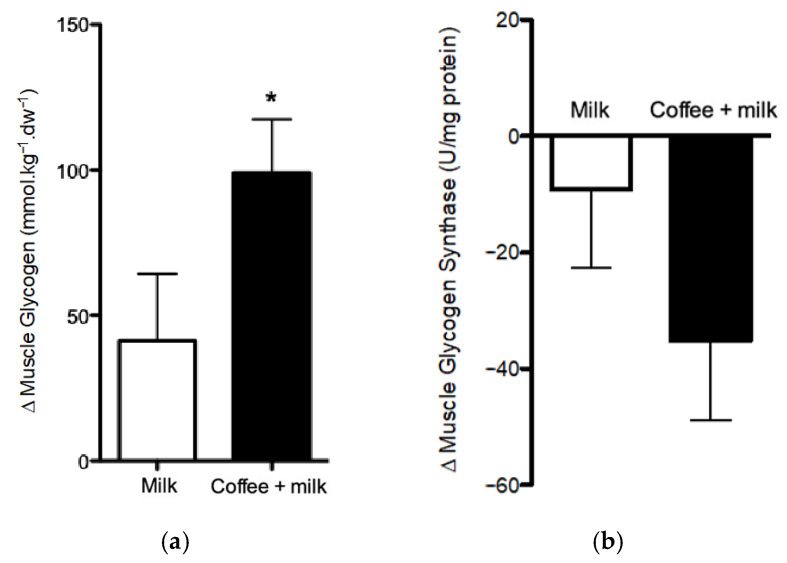
Muscle glycogen content (**a**) and Muscle glycogen synthase activity (**b**) in muscle of healthy endurance-trained adult men treated or not with coffee. Differences between 0 h and 4 h (Δ) following cycling to volitional fatigue (70% PPO). * *p* = 0.01 for muscle glycogen content (Coffee + milk Δ vs. Milk Δ). coffee + milk Δ vs. milk Δ for muscle glycogen synthase activity (*p* = 0.22). Data are least square means + SE from repeated procedure method, Proc Mixed (SAS studio).

**Figure 5 nutrients-13-03335-f005:**
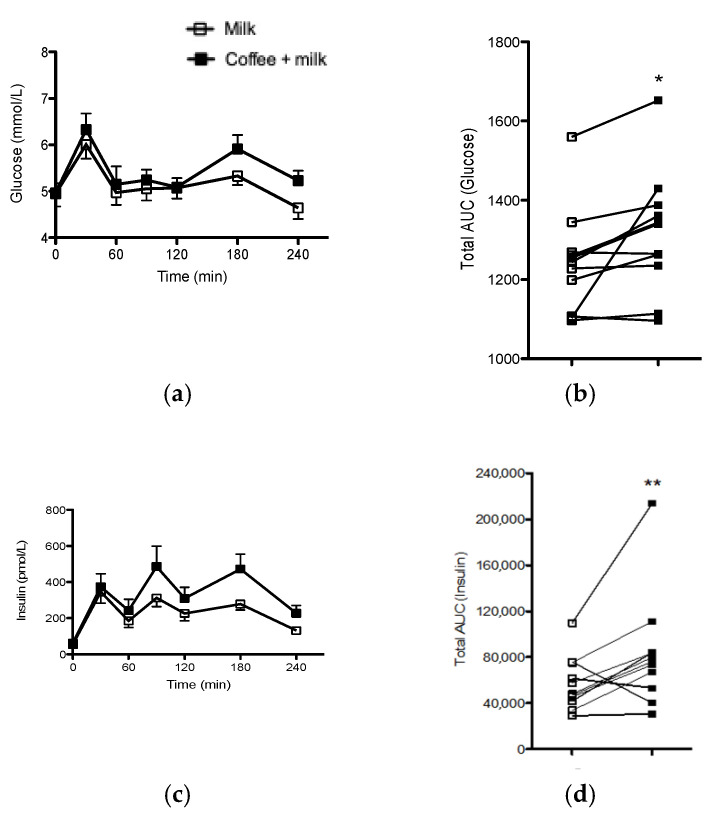
Glycemic (**a**) and insulinemic curve (**c**) in the seven time points during the 4-h recovery; individual glucose (**b**) and insulin (**d**) TAUC of healthy endurance-trained adult men treated or not with coffee. * *p* = 0.02 for glucose TAUC (coffee + milk vs. Milk); ** *p* = 0.03 for insulin TAUC (Coffee + milk vs. Milk). Repeated procedure method, Proc Mixed (SAS studio).

**Table 1 nutrients-13-03335-t001:** Exercise time and macronutrient intake in the evening meal for glycogen depletion in experimental sessions.

	Milk	Coffee + Milk
Total depleting exercise time (min)	116 ± 12.15	114 ± 12.15
Carbohydrates (g kg^−1^)	0.67 ± 0.05	0.62 ± 0.05
Proteins (g kg^−1^)	0.75 ± 0.07	0.80 ± 0.07
Lipids (g kg^−1^)	0.46 ± 0.07	0.50 ± 0.07

Data are means ± SE. Milk vs. Coffee + milk: *p* > 0.05 for all variables.

## Data Availability

The data presented in this study are available on request from the corresponding author. The data are not publicly available due to future planned analysis.

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
