# Peer review of "Coffee Increases Post-Exercise Muscle Glycogen Recovery in Endurance Athletes: A Randomized Clinical Trial"

_nutrients, 2021, doi:10.3390/nu13103335_

Round 1

Reviewer 1 Report

The manuscript concerns the effect of coffee consumption on the resynthesis of muscle glycogen and the glycemic and insulin response in people after exhausting exercise.

The manuscript is well written, the methodology has been thoroughly described and does not raise any objections. Although the number of people included in the overall experiment is rather small - only eleven persons.

The manuscript requires some minor corrections.

In the abstract, line 28, it is not known what the numbers in brackets refer to, the first one concerns the average age, the next one is probably BMI, the next one are difficult to guess.

Some shortcuts are weirdly assigned, for example CHO – Carbohydrates or PTN – Proteins.

Author Response

REVIEWER 1

COMMENT:

In the abstract, line 28, it is not known what the numbers in brackets refer to, the first one concerns the average age, the next one is probably BMI, the next one are difficult to guess.

RESPONSE:

Authors agree.

ACTION TAKEN:

We added the name or abbreviation of the variables.

L28: “Eleven participants were included in data analysis (age: 39.0 ± 6.0 years; BMI: 24.0 ± 2.3 kg / m2; VO2max: 59.9 ± 8.3 mL·kg-1·min-1; PPO: 346 ± 39 W)”.

COMMENT:

Some shortcuts are weirdly assigned, for example CHO – Carbohydrates or PTN – Proteins.

RESPONSE:

Authors agree.

ACTION TAKEN:

We changed CHO to carbohydrates and PTN to proteins throughout the text.

Reviewer 2 Report

The article is interesting and, in my opinion, it contributes a lot to the knowledge about post-workout regeneration.

The title of the article is consistent with its content. However, I would consider modifying it, because the research topic was the effect of coffee with the addition of CHO. What the authors emphasize in the Introduction (l.55-57). But this is only a slight suggestion.

Methodologically, the study was carried out correctly, but I noticed some errors in the statistical analysis and conclusions resulting from it. Details are provided below.

Discussion and Conclusions from the study are correct and interesting in my opinion. They broaden the knowledge about the influence of coffee on post-workout regeneration of the human body. In the discussion, at least in one place I noticed a rather questionable statement (details below). Maybe it is worth extending the Conclusions from the study? Your multi-day work ends with three sentences of conclusions.

Detailed notes:

  1. (l.62) What's with this coffee in Brazil? If you want to show how you like to drink coffee, write more about the favorite ways of brewing and drinking coffee in the region. And that the method of your research emerges from this favorite way of drinking coffee by Brazilians. Or don't write about it at all. This one sentence looks strange. Though it's interesting.
  2. (l.170-177) repeats the content (l.158-161) but the text is clearer and describes the meals better. Perhaps it is worth considering removing the description of meals from (l.158-161) or drastically shortening them to the statement "meals served".
  3. (l.176) You write that 8 mg of caffeine was given per kg of athlete's weight. How the served portions of coffee were converted into caffeine was not stated. The average caffeine content in coffee was used in general, in this particular type of coffee, did you analyze the coffee brewed in this aspect during the experiment?
  4. 4. (l.253-267) Methodology of statistical analyzes. You compare the mean values and report the value of the p parameter. The p parameter describes the significance in the statistical test. What test? It is worth writing what test was used in this case and what the hypotheses to which the p parameter refers to look like. In this case, I suppose, the t-test was used for two means from the dependent samples. The null hypothesis is the equality of the mean values in two populations (groups), the alternative hypothesis is the significant difference of the means. The smaller the p value, the greater the chance of the alternative hypothesis being true. The limit is your accepted significance of 0.05.
    Then in several places (l.298, 302, 309, ...) you give a confidence interval without describing what it refers to. Neither in the Methodology nor in the Discussion of the results. In the case of means, it is most likely the CI for the difference of means. In my opinion, this range does not show anything important. You don't comment on it anywhere in the text. I would remove it.
    You write further about the Cohen effect. And you give the confidence interval for the parameter d. What would this interval describe? Why do you put it in the text? Because you don't use it for any description. I understand that statistical programs count and record some strange things, but the researcher does not have to use all of them. Only those he deems useful in his study. These confidence intervals were useful for something?
  5. (l.305-309) In my opinion, you have an error in the interpretation of the obtained measures. You suggest that both treatments resulted decrease glycogen syntheze activity. The coffee therapy resulted in a greater decrease. Mathematically correct. But logically not anymore. As you wrote below, the synthesis of glycogen depends on its concentration in the blood. The lower the concentration, the greater the activity. It was not the therapies that caused the decrease in activity but the increase in concentration! Therapies do not cause the decline. There is no cause-and-effect relationship here. It can be said that in both therapies a decrease in activity was observed. Within a certain period of time.
    In particular, therapy with coffee did not cause a greater rate of decline in activity than therapy without coffee. There is another reason for this greater decline. Coffee caused a large increase in the synthesis activity, then the activity dropped to a level that was the same for both treatments. And in my opinion, this decrease in synthesis activity should not be interpreted in isolation from the higher peak level. Because we conclude that coffee reduces the activity of glycogen synthesis, and it contradicts your earlier conclusions. You should describe it better. Or remove.
  6. (l.324-329) The title of the Figure, in my opinion, should contain information about what the chart is about. And not about where the data came from and what the experiment leading to the obtained data looked like. I would remove the description of the experiment from the title of the Figure.
  7. (l.262-365) it was not the therapies that caused a decrease in activity, but "a decrease was observed in both therapies". All in all, there was no decline as there was a sharp increase first. And this decline in activity is a result of an earlier increase. What do you explain below in this paragraph.

Author Response

REVIEWER 2

COMMENT:

The title of the article is consistent with its content. However, I would consider modifying it, because the research topic was the effect of coffee with the addition of CHO. What the authors emphasize in the Introduction (l.55-57). But this is only a slight suggestion.

RESPONSE:

We appreciate the comment; however, we understand that coffee is the substance that differentiates the two treatments tested in this experiment. Therefore, the effect observed is due to the presence of coffee, not coffee+carbohydrate.

ACTION TAKEN:

None. We choose not to change the title.

COMMENT:

Discussion and Conclusions from the study are correct and interesting in my opinion. They broaden the knowledge about the influence of coffee on post-workout regeneration of the human body. In the discussion, at least in one place I noticed a rather questionable statement (details below). Maybe it is worth extending the Conclusions from the study? Your multi-day work ends with three sentences of conclusions.

RESPONSE:

Author agree with the suggestion of extending the conclusions from the study.

ACTION TAKEN:

We extended the study conclusion.

L578: “The consumption of coffee with sweetened milk improved the muscle glycogen resynthesis during the 4-h recovery period after exhaustive cycling exercise, when compared to the consumption of sweetened milk. The addition of coffee to a post-exercise beverage with adequate amounts of carbohydrates is an effective strategy to improve muscle glycogen recovery for those cycling athletes with a short-time recovery (< 4h) or competitions with multiple and sequential bouts of exercise. It is currently not possible to define the exact coffee components underlying the latter effect. However, caffeine, caffeic acid, and cafestol are plausible candidates, given previous findings indicating their effect on insulin secretion and muscle glucose uptake. These findings add to the current knowledge on the ergogenic properties of coffee.”

DETAILED NOTES:

COMMENT:

  1. (l.62) What's with this coffee in Brazil? If you want to show how you like to drink coffee, write more about the favorite ways of brewing and drinking coffee in the region. And that the method of your research emerges from this favorite way of drinking coffee by Brazilians. Or don't write about it at all. This one sentence looks strange. Though it's interesting.

RESPONSE:

Authors agree that the sentence was not adequate.

ACTION TAKEN:

We rewrote the sentence with more detailed information about the study that concluded about the coffee brewing and preparation method most used by Brazilians.

L64: “Moreover, a study conducted with data from the Brazilian National Dietary Survey collected from a probabilistic sample showed that the coffee brewing and preparation method most frequently used by Brazilians was filtered coffee and coffee with milk [14]”.

COMMENT:

  1. (l.170-177) repeats the content (l.158-161) but the text is clearer and describes the meals better. Perhaps it is worth considering removing the description of meals from (l.158-161) or drastically shortening them to the statement "meals served".

RESPONSE:

Authors agree with the suggestion of shortening the first description of the meals.

ACTION TAKEN:

We shortened the sentence.

L173: “After the procedure, the volunteer received three doses of test beverages (0, 60 and 120 min).”

COMMENT:

  1. (l.176) You write that 8 mg of caffeine was given per kg of athlete's weight. How the served portions of coffee were converted into caffeine was not stated. The average caffeine content in coffee was used in general, in this particular type of coffee, did you analyze the coffee brewed in this aspect during the experiment?

RESPONSE:

The average caffeine content in the specific coffee used in the experiment was obtained by laboratory analysis (Laboratory of Nutritional and Food Biochemistry, of the Department of Chemistry of the University Federal of Rio de Janeiro) and we considered a reference of 50% caffeine retention in the filter paper of coffee.

Reference for caffeine filter retention: Farah, A. Coffee Constituents. In Coffee: Emerging Health Effects and Disease Prevention; Chu, Y.-F., Ed.; Blackwell Publishing Ltd, 2012; pp. 22–58.

ACTION TAKEN:

We added more details about the coffee analysis and preparation.

L198: “To prepare the beverages and sandwiches, the ingredients were weighed according to the participants’ body mass. A content of 1530 mg of caffeine per 100g of coffee powder was obtained from the analysis of the coffee (Café Torrado e Moído Melitta TradicionalÒ) by HPLC, in the Laboratory of Nutritional and Food Biochemistry, of the Department of Chemistry of the University Federal of Rio de Janeiro. Coffee preparation was carried out using the 10% filtering method with boiling water in a filter paper (MelittaÒ 102), considering a 50% retention of caffeine in the filter [18] and respecting the ratio of 10g of powder to 100 mL of water.”

COMMENT:

  1. (l.253-267) Methodology of statistical analyzes. You compare the mean values and report the value of the p parameter. The p parameter describes the significance in the statistical test. What test? It is worth writing what test was used in this case and what the hypotheses to which the p parameter refers to look like. In this case, I suppose, the t-test was used for two means from the dependent samples. The null hypothesis is the equality of the mean values in two populations (groups), the alternative hypothesis is the significant difference of the means. The smaller the p value, the greater the chance of the alternative hypothesis being true. The limit is your accepted significance of 0.05.
    Then in several places (l.298, 302, 309, ...) you give a confidence interval without describing what it refers to. Neither in the Methodology nor in the Discussion of the results. In the case of means, it is most likely the CI for the difference of means. In my opinion, this range does not show anything important. You don't comment on it anywhere in the text. I would remove it. You write further about the Cohen effect. And you give the confidence interval for the parameter d. What would this interval describe? Why do you put it in the text? Because you don't use it for any description. I understand that statistical programs count and record some strange things, but the researcher does not have to use all of them. Only those he deems useful in his study. These confidence intervals were useful for something?

RESPONSE:

  1. a) We used the t-test. Authors agree with the suggestion of writing what test was used in this case and what the hypotheses to which the p parameter refers.
  2. b) The confidence interval presented with the p-value refers to the difference of means. Authors agree with the reviewer suggestion of removing it, once we showed the p-value and did not discuss the CI.
  3. c) Authors also decided to remove the CI for the parameter d because they do not add relevant

ACTION TAKEN:

  1. a) We completed the sentence of statistical analysis with the test we used and the hypotheses to which the p parameter refers.

L302: “The least-square means (LS means) values and their standard error (SE) of SAS PROC MIXED were presented and compared between Coffee + milk and Milk treatments using the Student’s t-test at a significance level of 0.05. The null hypothesis was the equality of the means values between the two treatments, and the alternative hypothesis was the significant difference of the means.”

  1. b) We removed the 95%CI, referring to the difference of means (L307, 347, 357, 364, 374-376).
  2. c) We removed the 95%CI, referring to the parameter d (L358, 376, 377).

COMMENT:

  1. (l.305-309) In my opinion, you have an error in the interpretation of the obtained measures. You suggest that both treatments resulted decrease glycogen syntheze activity. The coffee therapy resulted in a greater decrease. Mathematically correct. But logically not anymore. As you wrote below, the synthesis of glycogen depends on its concentration in the blood. The lower the concentration, the greater the activity. It was not the therapies that caused the decrease in activity but the increase in concentration! Therapies do not cause the decline. There is no cause-and-effect relationship here. It can be said that in both therapies a decrease in activity was observed. Within a certain period of time.

In particular, therapy with coffee did not cause a greater rate of decline in activity than therapy without coffee. There is another reason for this greater decline. Coffee caused a large increase in the synthesis activity, then the activity dropped to a level that was the same for both treatments. And in my opinion, this decrease in synthesis activity should not be interpreted in isolation from the higher peak level. Because we conclude that coffee reduces the activity of glycogen synthesis, and it contradicts your earlier conclusions. You should describe it better. Or remove.

RESPONSE:

Authors agree with the suggestion. Indeed, we cannot state a cause-and-effect relationship between coffee consumption and the reduction in glycogen synthase activity. Rather, the effect of coffee is more likely to be related to increased enzyme activity, leading to greater glycogen accumulation.

ACTION TAKEN:

We rewrote the paragraph to make it clear the idea about the effect of coffee on muscle glycogen synthase activity.

L456: “Considering the increase in glucose and insulin levels in response to the consumption of coffee, we suggest that coffee promoted greater muscle glucose uptake, with earlier activation of glycogen synthase and consequently earlier and greater glycogen resynthesis during the recovery period. As the high glycogen concentration inhibits the enzyme activation [31], the earlier and greater glycogen resynthesis was followed by the inhibition of glycogen sythase activity at the end of the recovery period (240 min). On the other hand, Milk treatment presumably resulted in later glycogen synthase maximal activity during recovery, which is consistent with the lower post-exercise muscle glycogen accumulation. Therefore, we cannot state a cause-and-effect relationship between coffee consumption and the reduction in glycogen synthase activity. Rather, the effect of coffee is more likely to have been due to an early increase of enzyme activity, leading to greater glycogen accumulation. Similar findings were reported by Battram et al (2004), who investigated the effects of caffeine intake on glycogen resynthesis in healthy men after glycogen-depleting exercise. The authors reported that caffeine intake was associated with a significant decrease in glycogen synthase activity at the end of the 5-h recovery period [33]. In our study, the high interpersonal variability in glycogen synthase enzymatic activity among the participants may explain the non-significant difference between treatments. Furthermore, a muscle biopsy performed at the mid-recovery time could have shown the increase in glycogen synthase activity promoted by coffee consumption before the reduction in activity promoted by the accumulation of muscle glycogen. It is important to point that we did not estimate sample size using enzyme activity variability data. Therefore, a larger sample size should be considered in future trials to define the effects of coffee consumption on glycogen synthase activity.”

COMMENT:

  1. (l.324-329) The title of the Figure, in my opinion, should contain information about what the chart is about. And not about where the data came from and what the experiment leading to the obtained data looked like. I would remove the description of the experiment from the title of the Figure.

RESPONSE:

Authors agree.

ACTION TAKEN:

We removed the description of the experiment from title of the figures 4 and 5 and added the participants characteristics.

L356: “Figure 4. Muscle glycogen content (a) and Muscle glycogen synthase activity (b) in muscle of healthy endurance-trained adult men treated or not with coffee. Differences between 0 h and 4 h (Δ) following cycling to volitional fatigue (70 % PPO). *p = 0.01 for muscle glycogen content (Coffee + milk Δ vs. Milk Δ). Coffee + milk Δ vs. Milk Δ for muscle glycogen synthase activity (p = 0.22). Data are least square means + SE from repeated procedure method, Proc Mixed (SAS studio).”

L395: “Figure 5. Glycemic (a) and insulinemic curve (c) in the seven time points during the 4-h recovery; individual glucose (b) and insulin (d) TAUC of healthy endurance-trained adult men treated or not with coffe. *p = 0.02 for glucose TAUC (Coffee + milk vs. Milk); **p = 0.03 for insulin TAUC (Coffee + milk vs. Milk). Repeated procedure method, Proc Mixed (SAS studio).”

COMMENT:

  1. (l.262-365) it was not the therapies that caused a decrease in activity, but "a decrease was observed in both therapies". All in all, there was no decline as there was a sharp increase first. And this decline in activity is a result of an earlier increase. What do you explain below in this paragraph.

RESPONSE:

Authors agree.

ACTION TAKEN:

We rewrote the sentence as suggested by the reviewer, because we understand the decrement in glycogen synthase activity was not due to different treatments, but to different muscle glycogen accumulation.

L360: “At the end of the recovery period, a decrease in glycogen synthase activity was observed with both treatments.”

EXTRA ACTION TAKEN:

To add a reference in the text we updated the reference list and citations throughout the text.
